# RETHINKING SEMANTIC FEW-SHOT IMAGE CLASSIFICATION

## ABSTRACT

Few-shot learning aims to train models that can be generalized to novel classes with only a few samples. Recently, a line of works has been proposed to enhance few-shot learning with semantic information from class names. However, these works focus on injecting semantic information into existing modules such as visual prototypes and feature extractors of the standard few-shot learning framework, which requires complex designs of the fusion mechanism. In this paper, we propose a novel few-shot learning framework that uses public textual encoders based on contrastive learning. To address the challenge of alignment between visual features and textual embeddings obtained from public textual encoders, we carefully design the textual branch of our framework and introduce a metric module to generalize the cosine similarity. For better transferability, we let the metric module adapt to different few-shot tasks and adopt MAML to train the model via bi-level optimization. Moreover, we conduct extensive experiments on multiple benchmarks to demonstrate the effectiveness of our method.

## 1 INTRODUCTION

Deep neural networks (Krizhevsky et al., 2017; Simonyan & Zisserman, 2014; Szegedy et al., 2015; He et al., 2016) have achieved remarkable success in many fields. However, training deep neural networks requires a large number of labeled data, which can be expensive and time-consuming to obtain. For instance, in medical imaging, obtaining labeled data requires expert radiologists to annotate images. This limits the application of deep learning models in real-world scenarios. In contrast, humans possess the ability to recognize and classify objects of unseen categories with only a few examples. This highlights the potential value of few-shot learning (Bart & Ullman, 2005; Fink, 2004; Fei-Fei et al., 2006; Lake et al., 2011), where models are trained on base classes and can be generalized well to novel classes with limited amounts of samples.

Previous works mainly focus on image classification tasks, and most of them adopt the meta-learning paradigm (Vinyals et al., 2016; Snell et al., 2017; Finn et al., 2017; Sung et al., 2018; Zhang et al., 2020). Recent works consider leveraging additional information from other modalities such as text to enhance the performance of few-shot learning. In particular, some methods (Xing et al., 2019; Peng et al., 2019; Li et al., 2020a) adopt static word embedding models (e.g., GloVe (Pennington et al., 2014)) to extract textual representations of class names and use them to adjust visual prototypes or classifiers. With the appearance of general language models (e.g., BERT (Devlin et al., 2018)) and vision-language pre-trained models (e.g., CLIP (Radford et al., 2021)), another line of works (Afham, 2022; Chen et al., 2023; Yang et al., 2023) adopt public pre-trained language models or textual encoders in vision-language models to extract more comprehensive semantic information from class names. However, these works still focus on injecting semantic information into existing modules (e.g., visual prototypes and feature extractors) of the standard few-shot learning framework, which requires complex designs of the fusion mechanism.

Inspired by the success of vision-language pre-trained models (Radford et al., 2021; Jia et al., 2021) trained by contrastive learning, we explore the idea of aligning visual features and textual embeddings for few-shot image classification in this paper, where textual embeddings are extracted by different types of public textual encoders from class names following the setting (Afham, 2022; Chen et al., 2023). However, there are two main factors making this alignment challenging. Firstly, unlike vision-language pre-trained models that have sufficient pairs of image and textual descriptions available for

model training, we only have the class name of each image instead of a rich description. Secondly, in contrast to vision-language pre-trained models where both visual and textual encoders are learnable to align embeddings, we utilize frozen public textual encoders. This leads to totally different structures of textual embedding spaces and thus makes the alignment between visual and textual features difficult, especially when using word embedding models and language models trained on uni-modal text data. For instance, if we directly align visual features and textual embeddings, the probability[1] of a sample image being assigned to its true label is extremely low (see blue bars in Figure 1). This indicates that the visual feature of an image is hard to approach the corresponding text embedding of its true label.

In this paper, we propose a novel framework (Figure 2) to boost few-shot learning by means of public textual encoders. To bridge the gap between visual and textual modalities, we carefully design a textual branch of our framework for three types of public textual encoders and introduce a metric module to measure the similarity between visual and textual embeddings. For masked language models, the textual branch first incorporates class labels into a prompt template containing a [MASK] token and then inputs the filled sentence to the textual encoder. The textual encoder transforms the input sentence into a hidden vector sequence and the final textual embedding is extracted from the vector corresponding to the [MASK] token. For static word embedding models and vision-language pre-trained models, we keep the same extraction method as (Chen et al., 2023). Meanwhile, the visual feature is obtained by a standard visual encoder. After that, we compute the similarities between visual features and textual embeddings

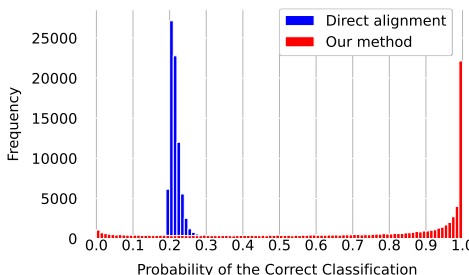

Figure 1: Frequency histogram of probability that each sample image is classified to true label. 80000 samples on novel classes of *mini*ImageNet dataset are collected with 5-way 5-shot setting. For direct alignment, we directly align visual features and textual embeddings with cosine similarity. The horizontal axis reflects the probability that each sample image is classified to its true label, which is output by the model. The vertical axis represents the total number of samples in each probability interval.

through the proposed metric module and send them into the contrastive loss. For better transferability on novel classes, we let the metric module adapt to different few-shot tasks and adopt Model-Agnostic Meta-Learning (MAML) (Finn et al., 2017) to train the model via bi-level optimization. Moreover, we conduct extensive experiments on multiple benchmarks to demonstrate that the proposed method significantly outperforms the state-of-the-art semantic few-shot learning methods.

The main contributions of this paper can be summarized as follows.

- We propose a novel few-shot learning framework that leverages semantic information extracted by a public textual encoder based on contrastive learning.

- We carefully design a textual branch of the framework and introduce a metric module to generalize the similarity measure.

- The metric module is designed to be adaptive to different few-shot tasks for better transferability, and MAML is adopted to train the model via bi-level optimization.

- We conduct extensive experiments on multiple benchmarks with different domains to demonstrate the effectiveness of our method.

## 2 RELATED WORK

**Few-shot Learning.** In general, few-shot learning methods are mainly divided into two categories: metric-based methods and optimization-based methods. Metric-based methods aim to map samples into an appropriate embedding space on the basis of certain distance metrics. Most previous methods use task-agnostic distance metrics, e.g., cosine similarity distance (Vinyals et al., 2016), Euclidean distance (Snell et al., 2017), CNN relation module (Sung et al., 2018), and Earth Mover's Distance

---

[1]Here probabilities mean the elements outputted by softmax function.

(Zhang et al., 2020). Additionally, several methods (Yoon et al., 2019; Li et al., 2019; Qiao et al., 2019; Ye et al., 2020; Simon et al., 2020; Akata et al., 2015) involve learning task-specific distance metrics, which can be adjusted for different tasks. Optimization-based methods (Finn et al., 2017; Rusu et al., 2018; Sun et al., 2019; Lee et al., 2019; Chen et al., 2021) aim to learn optimal initial model parameters on base classes and quickly fine-tune them on novel classes with a few support examples. Different from zero-shot learning (Akata et al., 2015), we use MAML (Finn et al., 2017) to optimize the metric module with the support set and make it task-specific on the query set. Besides, inductive FSL(e.g., (Vinyals et al., 2016), (Snell et al., 2017)) and transductive FSL (e.g., (Singh & Jamali-Rad, 2022), (Hu et al., 2023), (Hu et al., 2022)) are two different branches of FSL. The former predicts query samples one by one while the latter predicts query samples as a whole. Our work belongs to the inductive FSL.

**Few-shot Learning with Semantic Information.** Recent works (Xing et al., 2019; Peng et al., 2019; Li et al., 2020a; Yan et al., 2021; Afham, 2022; Yang et al., 2023; Chen et al., 2023) on few-shot learning start to utilize semantic information from class labels to enhance few-shot learning. AM3 (Xing et al., 2019) proposes an adaptive modality mixture mechanism to model prototype representation as a combination of visual features and language semantic features. KTN (Peng et al., 2019) learns classifiers by fusing visual information and knowledge information acquired from a knowledge graph and word embeddings with a semantic-visual mapping network based on Graph Convolutional Network (Kipf & Welling, 2016). VS-Alignment (Afham, 2022) introduces a contrastive alignment between visual and semantic features as an additional objective. Xiao et al. (2022) utilize category name embeddings from a vision-language model to initialize the classification head. Semantic Prompt (Chen et al., 2023) considers semantic information as prompts to tune the ViT (Dosovitskiy et al., 2020) feature extractor. In contrast to these works, we propose a new few-shot learning framework to align visual and textual embeddings via contrastive learning. Different from VS-Alignment, we introduce a task-specific image-text metric module, which can adaptively align the visual and textual features, leading to substantial performance improvement.

**Contrastive Learning.** Contrastive learning is a popular method in self-supervised representation learning. It learns representations by pulling positive samples close and driving negative samples away from them in the latent embedding space with a contrastive loss. A set of previous works have shown the excellent performance of contrastive learning in computer vision (He et al., 2020; Chen et al., 2020b;a; Yang et al., 2022) and natural language processing (Liu & Sun, 2015; Huang et al., 2018; Lee et al., 2020b) tasks. Furthermore, recent works (Zhang et al., 2022; Radford et al., 2021; Jia et al., 2021; Alayrac et al., 2022; Yuan et al., 2021; Afham, 2022; Yang et al., 2022) apply contrastive learning to multi-modal settings by aligning image-text pairs in the embedding space. Different from previous multimodal contrastive learning works, our work introduces contrastive learning to few-shot learning and proposes a learnable metric module to make aligning visual features and textual embeddings possible.

## 3 PROBLEM DEFINITION

Few-shot learning involves two disjoint class sets: a base class set $\mathcal{C}_{base}$ classes and a novel class set $\mathcal{C}_{novel}$ classes. Sufficient labeled samples are provided for each base class, while abundant unlabeled samples and only a few labeled samples are provided for each novel class. Few-shot learning targets at classifying unlabeled samples from novel classes through training on all the given labeled samples. Previous works usually formulate the few-shot learning problem as $N$-way $K$-shot classification, which denotes a classification task among $N$ classes with $K$ labeled samples available for each class. In addition, given a fixed public textual encoder, we use bimodal contrastive learning to leverage the semantic information extracted by it. Concretely, for each embedded sample image $z$ and $N$ embedded class labels $\{t_1, t_2, \ldots, t_N\}$ in a $N$-way $K$-shot classification task, contrastive learning adjusts the embedding space through the following widely-used contrastive loss (Oord et al., 2018; Chen et al., 2020a; He et al., 2020; Chen et al., 2020b) (using cosine similarity as an example):

$$\mathcal{L} = -\log \frac{\exp(z \cdot t_+/\tau)}{\sum_{i=1}^{N} \exp(z \cdot t_i/\tau)}, \tag{1}$$

where $t_+$ is the embedded true label of the sample image and $\tau$ is a temperature hyper-parameter.

Meta-learning paradigm (Vinyals et al., 2016; Finn et al., 2017) is commonly used to solve the few-shot learning problem, which trains and evaluates the model with the episodic mechanism. The

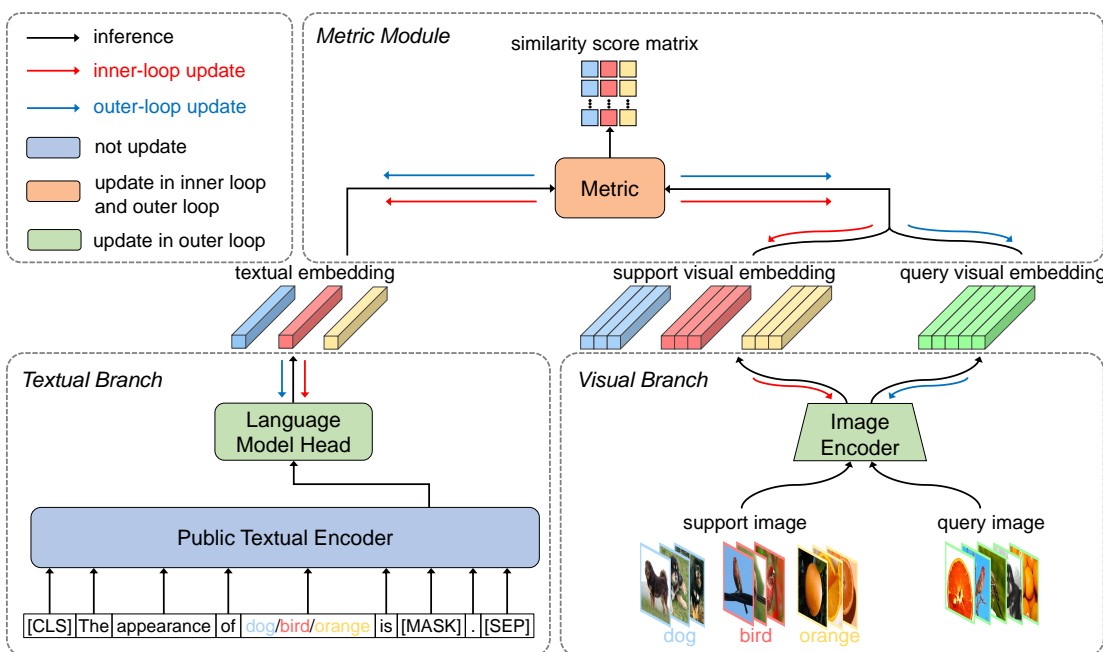

Figure 2: The overview of our framework. For each episode, class labels are fed into the textual branch to obtain the textual embeddings. The support visual embeddings and query visual embeddings are extracted by the visual branch from support and query images respectively. To align the visual and textual embeddings, we propose a metric module to generalize the similarity measure and output the similarity score matrix. Moreover, for better transferability, we let the metric module be adaptive to different few-shot tasks via bi-level optimization.

standard meta-learning paradigm contains two stages: meta-training and meta-testing. In each episode of the meta-training stage, a $N$-way $K$-shot $M$-query classification task $\mathcal{T} = (\mathcal{S}, \mathcal{Q})$ is constructed with samples from the base classes. We first randomly select $N$ classes from $\mathcal{C}_{base}$ as $\mathcal{C}_{\mathcal{T}}$. For each class, we randomly sample $K$ support images and $M$ query images. Then we form the support set $\mathcal{S} = \{(x_i, y_i) \mid y_i \in \mathcal{C}_{\mathcal{T}}, i = 1, 2, \ldots, N \times K\}$ and the query set $\mathcal{Q} = \{(x_i, y_i) \mid y_i \in \mathcal{C}_{\mathcal{T}}, i = 1, 2, \ldots, N \times M\}$ with the support images and the query images respectively, where $x_i$ is the $i$-th sample image and $y_i$ is the class label of $x_i$. To learn an appropriate embedding space, bi-level optimization is performed on $\mathcal{S}$ and $\mathcal{Q}$ respectively, utilizing a contrastive loss. In each episode of the meta-testing stage, a classification task is built on the novel classes in a similar way. The support set is formed with a few label samples, while the query set is sampled from the unlabeled samples. After adapting to the novel classes by minimizing the contrastive loss on the support set, the model is used to predict class labels for the sample images in the query set.

## 4 THE PROPOSED METHOD

We introduce our method of Few-shot Image classification with public Textual Encoders (FITE) in this section. The overall framework is illustrated in Figure 2, which consists of three modules: a textual branch, a visual branch, and a metric module. For each episode, the textual branch extracts textual embeddings from class labels, while the visual branch extracts visual embeddings from support and query images. Moreover, the metric module computes the similarity score matrix between textual and visual embeddings from these two branches. In addition, we utilize a training strategy based on MAML algorithm to train the model via bi-level optimization.

### 4.1 TEXTUAL BRANCH

In this section, we explain how we design the textual branch to get textual embeddings from class labels. The textual branch comprises a public textual encoder and a language model head. During

meta-training and meta-testing, the textual encoder is frozen while the language model head is tuned for the downstream classification tasks.

In our study, we adopt three types of public textual encoders: static word embedding models, masked language models, and vision-language pre-trained models. For static word embedding models, we take the average of the output vectors for each word in a class name as the textual embedding. For masked language models, we notice that they mainly take sentences rather than single words or phrases as input during the pre-training stage. Therefore, to bridge the gap between the pre-training and downstream tasks, for each class label $y_i$, we insert it into a hand-crafted prompt template and get $y_i^{prompt}$ as the input of the textual encoder. The token sequence of $y_i^{prompt}$ is first converted to a token embedding sequence through a token vocabulary. The input embedding sequence is calculated by summing the corresponding token embeddings and positional embeddings. Then, the masked language model transforms the input embeddings into a sequence of hidden vectors. Two straightforward ways to get the textual embedding from the output hidden vector sequence are respectively: (1) taking the average vector of the output vector sequence as the textual embedding; (2) taking the hidden vector of the [CLS] token as the textual embedding. To make textual embeddings more relevant to the visual descriptive information of the corresponding categories, we design a prompt template with one [MASK] token as

$$y_i^{prompt} = [\text{CLS}] \text{ The appearance of } y_i \text{ is [MASK] . [SEP]}$$

and extract the textual embedding by sending the hidden vector of the [MASK] token to the language model head. In this way, the extraction of textual embeddings is treated as a masked language modeling task, which makes downstream classification tasks more consistent with the pre-training of the masked language model. For vision-language pre-trained models, we follow (Radford et al., 2021) and use "A photo of a $y_i$" as the prompt template, then extract textual embeddings from the sentence-initial token. Besides, we make the prompt templates learnable to avoid time-consuming prompt engineering following (Zhou et al., 2022). Specifically, the word embedding of each token in the prompt template is replaced by a learnable vector of the same dimension. These learnable vectors are initialized with original word embeddings and updated in the outer loop together with other parameters.

## 4.2 METRIC MODULE

Inspired by vision-language pre-trained models trained by contrastive learning, we explore aligning visual and textual modalities for few-shot image classification. However, directly aligning visual features and textual embeddings extracted by public textual encoders with cosine similarity has a poor effect in few-shot setting. The blue bars in Figure 1 show that the probability of a sample image being assigned to its true label is extremely low if we directly align the visual and textual embeddings. In this paper, we introduce a metric module to generalize the similarity measure between visual features and textual embeddings. Moreover, we let the metric module adapt to different few-shot tasks for better transferability on novel classes.

Specifically, we define $f_{\theta_I}$ as the image encoder with learnable parameters $\theta_I$ to transform each sample image $x_i$ into a feature map $z_i = f_{\theta_I}(x_i)$. Textual branch $f_{\theta_T}$ with learnable parameters $\theta_T$ is used to extract the textual embedding $t_{y_i} = f_{\theta_T}(y_i)$ from each class label $y_i$. We generalize the similarity measure between visual embeddings $z$ and textual embeddings $t$ as a learnable function $M(z, t)$ called metric module, whose parameters are denoted as $\theta_M$. For example, the metric module could be a bilinear function $M(z, t) = z^\top \theta_M t$ (degenerating to the cosine similarity if $\theta_M$ is the identity matrix) or a neural network, e.g., $M(z, t) = \text{MLP}_{\theta_M}([z, t])$. During meta-testing, we first fine-tune the task-specific parameters $\theta_M$ on the support set $\mathcal{S}$. Then we use the similarity score matrix computed by the metric module as a reference to infer labels for sample images in the query set $\mathcal{Q}$. As is shown in Figure 1, the correct classification probabilities of our method are significantly higher than that of direct alignment, which means that our metric module can effectively align the visual features and textual embeddings.

## 4.3 LOSS FUNCTION

We formulate the learning objective as a contrastive loss (Eq equation 1), which pulls together images and corresponding class labels while pushing away unmatched pairs in the embedding space.

Moreover, we aim to train a model to maximize the similarity between visual features and textual embeddings for matching (image, text) pairs while reducing the similarity for non-matching pairs. Specifically, for a classification task $\mathcal{T} = (\mathcal{S}, \mathcal{Q})$, we calculate the contrastive loss on the support set $\mathcal{S}$ and the query set $\mathcal{Q}$ respectively. On the support set, the contrastive loss $\mathcal{L}_{\mathcal{S}}$ is computed with all the support samples, which has a formulation as:

$$\mathcal{L}_{\mathcal{S}} = -\frac{1}{|\mathcal{S}|} \sum_{x_i \in \mathcal{S}} \log \frac{\exp\left(M(z_i, t_{y_i})/\tau\right)}{\sum_{c \in \mathcal{C}_{\mathcal{T}}} \exp\left(M(z_i, t_c)/\tau\right)}, \tag{2}$$

where $z_i$ is the visual embedding of the $i^{th}$ support image $x_i$, $t_{y_i}$ is the textual embedding of the true label $y_i$ corresponding to $x_i$, $t_c$ is the textual embedding of the class label $c$, and $M(\cdot, \cdot)$ is the similarity measure. On the query set, the contrastive loss $\mathcal{L}_{\mathcal{Q}}$ has almost the same formulation as $\mathcal{L}_{\mathcal{S}}$, except it is computed with all the query samples of $\mathcal{Q}$.

### 4.4 TRAINING STRATEGY

In this work, we incorporate the Model-Agnostic Meta-Learning (MAML) (Finn et al., 2017) algorithm to train the model via bi-level optimization as our training strategy. Our training strategy aims to learn a good model initialization (through the outer-loop optimization), which can be quickly adapted to novel tasks given a few examples (through the inner-loop optimization). The whole algorithm for our training strategy is outlined in Algorithm 1.

First, we initialize the parameters of image encoder $\theta_I$, language model head $\theta_T$, and metric module $\theta_M$. For each task instance $\mathcal{T}_j$ from the distribution $p(\mathcal{T})$, we divide $\mathcal{T}_j$ into a support set $\mathcal{S}_j$ and a query set $\mathcal{Q}_j$. Intuitively, feature extraction is general for downstream classification tasks. However, image-text alignment, which is conducted by the metric module, is task-specific as each concrete classification task focuses on different parts of features. To let the metric module be task-specific, we create copies of $\theta_M$ as the adapted parameters $\theta'_M$. In the inner loop, we adapt the model to the current task $\mathcal{T}_j$ by updating $\theta'_M$ with a number of gradient descent steps on the support set while keeping $\theta_I$, $\theta_T$ and $\theta_M$ fixed. In the outer loop, $\theta'_M$ are utilized to evaluate the performance of the adapted model on the query set. Specifically, we compute loss on the query set with $\theta_I$, $\theta_T$, $\theta'_M$ and perform gradient descent with respect to all the model parameters $\theta = \{\theta_I, \theta_T, \theta_M\}$. The optimization objective of the meta-training stage is to learn a good initialization across tasks. For example, when using one gradient update in the inner loop, the optimization objective can be formulated as follows:

$$\min_{\theta} \sum_{\mathcal{T}_j \sim p(\mathcal{T})} \mathcal{L}_{\mathcal{Q}_j}(\theta_I, \theta_T, \theta_M - \alpha \nabla_{\theta_M} \mathcal{L}_{\mathcal{S}_j}(\theta_I, \theta_T, \theta_M)),$$

where $\mathcal{L}_{\mathcal{S}_j}$ and $\mathcal{L}_{\mathcal{Q}_j}$ denote the loss functions that evaluate the performance on support and query set respectively, and $\alpha$ is the learning rate of the inner loop.

## 5 EXPERIMENTS

### 5.1 SETUP

**Datasets.** We experiment on two general object recognition datasets, i.e., *mini*ImageNet, *tiered*ImageNet, and one fine-grained categorization image classification dataset, i.e., CUB-200-2011. The *mini*ImageNet dataset is proposed in (Vinyals et al., 2016) as a benchmark for few-shot image classification tasks. It contains a subset of 100 classes in the ImageNet (Russakovsky et al., 2015) dataset, where 64 classes are used for training, 16 classes for validation, and 20 classes for testing. The *tiered*ImageNet dataset (Ren et al., 2018), which is also derived from the ImageNet (Russakovsky et al., 2015) dataset, contains 351 classes for training, 97 classes for validation, and 160 classes for testing. CUB-200-2011 (CUB) (Wah et al., 2011) is a dataset for fine-grained bird species classification tasks, containing 100 classes for training, 50 classes for validation, and 50 classes for testing. We also evaluate the domain transferability of our method by training on *mini*ImageNet dataset and then testing on CUB dataset.

**Implementation.** For the visual branch, following previous works (Oreshkin et al., 2018; Lee et al., 2019), we use ResNet-12 as our image encoder of the visual branch. We apply a global average

---

**Algorithm 1:** Training strategy for our method

---

**Input:** Task distribution $p(\mathcal{T})$, learning rate $\alpha, \beta$.
**Output:** Model parameters $\theta$.
Initialize the parameters of image encoder $\theta_I$ with pre-trained model;
Initialize the parameters of language model head $\theta_T$, metric module $\theta_M$;
**while** *not done* **do**

> Sample a task instance $\mathcal{T}_j \sim p(\mathcal{T})$;
> Let $\mathcal{T}_j = (\mathcal{S}_j, \mathcal{Q}_j)$ ;
> Initialize adapted parameters of metric module $\theta'_M = \theta_M$;
> **for** *number of adaptation steps* **do**
>
> > Compute loss on the support set $\mathcal{L}_{\mathcal{S}_j}(\theta_I, \theta_T, \theta'_M)$ using Eq equation 2;
> > Update $\theta'_M \leftarrow \theta'_M - \alpha\nabla_{\theta'_M}\mathcal{L}_{\mathcal{S}_j}(\theta_I, \theta_T, \theta'_M)$;
>
> **end**
> Compute loss on the query set $\mathcal{L}_{\mathcal{Q}_j}(\theta_I, \theta_T, \theta'_M)$;
> Let $\theta = \{\theta_I, \theta_T, \theta_M\}$;
> Update $\theta \leftarrow \theta - \beta\nabla_\theta\mathcal{L}_Q(\theta_I, \theta_T, \theta'_M)$;

**end**

---

pooling layer after the last residual block. The backbone network takes images with a spatial size of $84 \times 84$ as input and outputs 640-dim support and query visual embeddings. To extract comprehensive semantic information from class names, we adopt three types of public textual encoders: GloVe (Pennington et al., 2014) as the static word embedding model, RoBERTa-base (Liu et al., 2019) as the masked language model and CLIP (Radford et al., 2021) as the vision-language pre-trained model. The language model head is a linear layer that transforms the output hidden vectors into 640-dim textual embeddings. We use the bilinear form of our metric module and initialize it as an identity matrix. More implementation details are provided in the Appendix.

## 5.2 COMPARISON WITH STATE-OF-THE-ART

**General Object Recognition and Fine-Grained Categorization.** For fair comparisons, we compare the results with other methods using the same backbone or similar methods for both 5-way 1-shot and 5-way 5-shot settings on *mini*ImageNet, *tiered*ImageNet and CUB-200-2011 datasets. As shown in Table 1, FITE-CLIP is superior to existing methods and achieves the best performance. Compared with previous methods that leverage semantic information from class names, such as KTN (Peng et al., 2019), AM3 (Xing et al., 2019), TRAML (Li et al., 2020a), VS-Alignment (Afham, 2022), LPE (Yang et al., 2023) and SP (Chen et al., 2023), our method improves the accuracy by 2.17% and 1.03% at 1-shot and 5-shot settings on *mini*ImageNet dataset respectively. Furthermore, FITE-CLIP outperforms SP-CLIP by 1.17% at 1-shot setting on *tiered*ImageNet dataset. FITE-GloVe and FITE-RoBERTa can achieve comparable performance with SP-GloVe and SP-SBERT, but without requiring RepeatAug (Berman et al., 2019). For fine-grained categorization image classification, as shown in Table 2, our method surpasses all the competitors on CUB dataset, including RE-Net (Kang et al., 2021), DeepEMD (Zhang et al., 2020) and LPE-CLIP, which previously achieved the best result. FITE-CLIP outperforms LPE-CLIP by 4.56% and 3.18% at 1-shot and 5-shot settings respectively. These results validate the effectiveness of our methods on the fine-grained categorization dataset.

**Evaluation on Cross Domain and Larger Shots.** To evaluate the cross-domain transferability of different few-shot learning methods, we train them on the source domain *mini*ImageNet dataset and test them on the target domain CUB dataset. This setting is challenging due to the domain gap between the training and testing datasets. The results are reported in Table 3, showing that our method has competitive performance and obtains consistent improvements in the cross-domain setting. This indicates the transferability of our method in a situation where the meta-testing tasks are entirely different from the meta-training tasks. Furthermore, we evaluate the performance when the number of shots increases (e.g., 10-shot, 30-shot, and 50-shot) in Table 4. This shows that our method would be more effective when there are more (image, text) pairs available for novel classes. These comparisons

| Method | Backbone | *mini*ImageNet | | *tiered*ImageNet | |
|---|---|---|---|---|---|
| | | 1-shot | 5-shot | 1-shot | 5-shot |
| MAML[†] (Finn et al., 2017) | ResNet-12 | 62.90±0.20 | 80.81±0.14 | 59.08±0.20 | 80.04±0.16 |
| CC (Chen et al., 2019) | ResNet-12 | 55.43±0.81 | 77.18±0.61 | 61.49±0.91 | 82.37±0.67 |
| MetaOptNet (Lee et al., 2019) | ResNet-12 | 62.64±0.61 | 78.63±0.46 | 65.99±0.72 | 81.56±0.53 |
| Meta-Baseline (Chen et al., 2021) | ResNet-12 | 63.17±0.23 | 79.26±0.17 | 68.62±0.27 | 83.74±0.18 |
| ProtoNet (Snell et al., 2017) | ResNet-12 | 62.39±0.20 | 80.53±0.20 | 68.23±0.23 | 84.03±0.16 |
| ConvNet (Wertheimer, 2019) | ResNet-12 | 64.59±0.45 | 82.02±0.29 | 69.75±0.52 | 84.21±0.26 |
| Rethink-Distill (Tian et al., 2020) | ResNet-12 | 64.82±0.60 | 82.14±0.43 | 71.52±0.69 | 86.03±0.49 |
| FEAT (Ye et al., 2020) | ResNet-12 | 66.78±0.20 | 82.05±0.14 | 70.80±0.23 | 84.79±0.16 |
| BML (Zhou et al., 2021) | ResNet-12 | 67.04±0.63 | 83.63±0.29 | 68.99±0.50 | 85.49±0.34 |
| RE-Net (Kang et al., 2021) | ResNet-12 | 67.60±0.44 | 82.58±0.30 | 71.61±0.51 | 85.28±0.35 |
| CAN (Hou et al., 2019) | ResNet-12 | 67.19±0.55 | 80.64±0.35 | 73.21±0.58 | 84.93±0.38 |
| SEMAN-G (Huang et al., 2022) | ResNet-12 | 68.24±0.82 | 83.48±0.48 | 71.06±0.92 | 86.02±0.58 |
| DeepEMD (Zhang et al., 2020) | ResNet-12 | 65.91±0.82 | 82.41±0.56 | 71.16±0.87 | 86.03±0.58 |
| TPMM (Wu et al., 2021) | ResNet-12 | 67.64±0.63 | 83.44±0.43 | 72.24±0.70 | 86.55±0.63 |
| ADM (Li et al., 2020b) | ResNet-12 | 65.87±0.43 | 82.05±0.29 | 70.78±0.52 | 85.70±0.43 |
| CL (Yang et al., 2022) | ResNet-12 | 70.19±0.46 | 84.66±0.29 | 72.62±0.51 | 86.62±0.33 |
| PAL (Ma et al., 2021) | ResNet-12 | 69.37±0.64 | 84.40±0.44 | 72.25±0.72 | 86.95±0.47 |
| SUN (Dong et al., 2022) | Visformer-S | 67.80±0.45 | 83.25±0.30 | 72.99±0.50 | 86.74±0.33 |
| KTN (Peng et al., 2019) | ResNet-12 | 61.42±0.72 | 74.16±0.56 | – | – |
| AM3 (Xing et al., 2019) | ResNet-12 | 65.30±0.49 | 78.10±0.36 | 69.08±0.47 | 82.58±0.31 |
| TRAML (Li et al., 2020a) | ResNet-12 | 67.10±0.52 | 79.54±0.60 | – | – |
| DeepEMD-BERT (Yan et al., 2021) | ResNet-12 | 67.03±0.79 | 83.68±0.65 | 73.67±0.72 | 87.51±0.75 |
| VS-Alignment (Afham, 2022) | ResNet-12 | 65.89±0.80 | – | – | – |
| OB Baran, et al. (Baran & Cinbis, 2022) | ResNet-12 | 69.76±0.21 | 81.19±0.18 | 72.69±0.20 | 85.29±0.17 |
| LPE-GloVe (Yang et al., 2023) | ResNet-12 | 68.28±0.43 | 78.88±0.33 | 72.03±0.49 | 83.76±0.37 |
| LPE-CLIP (Yang et al., 2023) | ResNet-12 | 71.64±0.40 | 79.67±0.32 | 73.88±0.48 | 84.88±0.36 |
| SP-GloVe (Chen et al., 2023) | Visformer-T | 70.81±0.42 | 83.31±0.30 | 74.68±0.50 | 88.64±0.31 |
| SP-SBERT (Chen et al., 2023) | Visformer-T | 70.70±0.42 | 83.55±0.30 | 73.31±0.50 | 88.56±0.32 |
| SP-CLIP (Chen et al., 2023) | Visformer-T | 72.31±0.40 | 83.42±0.30 | 78.03±0.46 | 88.55±0.32 |
| FITE-GloVe (Ours) | ResNet-12 | 70.39±0.58 | 83.57±0.40 | 73.43±0.68 | 87.27±0.45 |
| FITE-RoBERTa (Ours) | ResNet-12 | 70.43±0.61 | 83.90±0.40 | 73.52±0.71 | 87.67±0.45 |
| **FITE-CLIP (Ours)** | ResNet-12 | **74.48±0.55** | **84.45±0.40** | **79.20±0.43** | **88.85±0.13** |

Table 1: Comparison with previous works on *mini*ImageNet and *tiered*ImageNet. The result with [†] is reported in (Ye & Chao, 2021). Methods in the top rows do not use semantic information, and methods in the middle rows leverage semantic information from class names or descriptions. Accuracies are reported with 95% confidence intervals.

| Method | Backbone | 1-shot | 5-shot |
|---|---|---|---|
| MAML (Finn et al., 2017) | Conv-4 | 68.42±1.07 | 83.47±0.62 |
| CSS (An et al., 2021) | Conv-4 | 66.01±0.90 | 81.84±0.59 |
| SLA (Lee et al., 2020a) | Conv-4 | 45.94±0.87 | 68.62±0.75 |
| CC (Chen et al., 2019) | ResNet-12 | 67.30±0.86 | 84.75±0.60 |
| RelationNet (Sung et al., 2018) | ResNet-12 | 68.58±0.94 | 84.05±0.56 |
| ProtoNet (Snell et al., 2017) | ResNet-12 | 66.09±0.92 | 82.50±0.58 |
| FEAT (Ye et al., 2020) | ResNet-12 | 68.87±0.22 | 82.90±0.15 |
| RE-Net (Kang et al., 2021) | ResNet-12 | 79.49±0.44 | 91.11±0.24 |
| BML (Zhou et al., 2021) | ResNet-12 | 76.21±0.63 | 90.45±0.36 |
| DeepEMD (Zhang et al., 2020) | ResNet-12 | 75.65±0.83 | 88.69±0.50 |
| AM3-ProNet [†] (Yan et al., 2021) | ResNet-12 | 77.03±0.85 | 87.20±0.70 |
| LPE-CLIP (Yang et al., 2023) | ResNet-12 | 80.76±0.40 | 88.98±0.26 |
| FITE-GloVe (Ours) | ResNet-12 | 80.83±0.42 | 91.56±0.34 |
| FITE-RoBERTa (Ours) | ResNet-12 | 80.46±0.59 | 91.83±0.34 |
| **FITE-CLIP (Ours)** | ResNet-12 | **85.32±0.52** | **92.16±0.31** |

Table 2: Comparison with previous works on CUB-200-2011 dataset. The result with [†] is reported in (Yan et al., 2021). Methods in the top rows do not use semantic information, and methods in the middle rows leverage semantic information. Accuracies are reported with 95% confidence intervals.

demonstrate that our method has a more robust transferability, which means it can work well in cross-domain and larger shots scenarios.

| Method | miniImageNet → CUB |
|---|---|
| MAML (Finn et al., 2017) | 51.34±0.72 |
| ProtoNet (Snell et al., 2017) | 62.02±0.70 |
| Rethink-Distill (Tian et al., 2020) | 68.57±0.39 |
| Centroid (Afrasiyabi et al., 2020) | 70.37±1.02 |
| **FITE-GloVe (Ours)** | **73.84±0.54** |

Table 3: Cross-domain comparison on CUB-200-2011 dataset in a 5-way 5-shot setting with 95% confidence intervals.

| Method | 10-shot | 30-shot | 50-shot |
|---|---|---|---|
| SimpleShot (Wang et al., 2019) | 84.89 | 87.53 | 88.08 |
| AM3 (Xing et al., 2019) | 81.57 | – | – |
| ProtoNet (Snell et al., 2017) | 82.83 | 85.07 | 85.57 |
| FEAT (Ye et al., 2020) | 85.15 | 87.82 | 87.83 |
| **FITE-RoBERTa (Ours)** | **86.86** | **88.92** | **90.59** |

Table 4: 5-way 10/30/50-shot classification accuracies on miniImageNet over 1000 tasks with 95% confidence intervals.

| Metric Module | GloVe | | RoBERTa | | CLIP | |
|---|---|---|---|---|---|---|
| | 1-shot | 5-shot | 1-shot | 5-shot | 1-shot | 5-shot |
| ✗ | 62.08±0.58 | 74.38±0.54 | 62.97±0.93 | 75.30±0.56 | 66.15±0.60 | 76.41±0.59 |
| ✓ | **70.39±0.58** | **83.57±0.40** | **70.43±0.61** | **83.90±0.40** | **74.48±0.55** | **84.45±0.40** |

Table 5: Ablation study of the metric module on miniImageNet dataset. "✗" means that we remove the metric module. "✓" means that we use our metric module to train the model.

## 5.3 ABLATION STUDY

**Ablation study of the metric module.** The ablation study results on the metric module are shown in Table 5. To verify the effectiveness of the metric module, we remove it and conduct experiments for three types of textual encoders. The results show a significant decrease in performance of three methods on miniImageNet dataset, demonstrating the importance of the task-specific metric module. By leveraging the metric module to generalize the cosine similarity, our model can adaptively measure the similarity between visual features and textual embeddings for different few-shot tasks.

| Backbone | miniImageNet | | tieredImageNet | |
|---|---|---|---|---|
| | 1-shot | 5-shot | 1-shot | 5-shot |
| ResNet-12 | 74.48±0.55 | 84.45±0.40 | 79.20±0.43 | 88.85±0.13 |
| ViT-Small | 74.82±0.76 | 86.19±0.54 | 81.38±0.88 | 89.53±0.61 |
| Swin-Tiny | **76.49±0.75** | **87.33±0.54** | **81.98±0.80** | **90.48±0.54** |

Table 6: Comparison among FITE-CLIP with different visual backbones.

**Ablation study of the visual backbone.** To study the dependence on the visual backbone, we also conduct experiments with ViT-Small(Dosovitskiy et al., 2020) and Swin-Tiny(Dosovitskiy et al., 2014) as the visual backbone on miniImageNet and tieredImageNet. From the results shown in Table 6, we observe that the replacement of the backbone from ResNet-12 with ViT-Small and Swin-Tiny yields significant performance improvements. The results also suggest that our approach can adapt well to different types of backbones without the need for adjustments.

## 6 CONCLUSION

In this paper, we propose a novel few-shot learning framework with public text encoders to boost few-shot learning. To address the difficulty in aligning visual and textual features in the few-shot setting, we introduce a task-specific metric module and design a meta-learning framework, obtaining models with strong performance using different types of language models. The success of our approach suggests that alignment offers a more promising approach compared to feature fusion in semantic few-shot learning.

## REPRODUCIBILITY STATEMENT

**(This section does not count towards the page limit.)**

We provide the detailed experimental implementation details in the Appendix. We will make our codes and checkpoints publicly available to facilitate the replication and verification of our results upon publication.

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

# Appendix

## A    IMPLEMENTATION DETAILS

We first pre-train the image encoder for 200 epochs on *mini*ImageNet and 100 epochs on *tiered*ImageNet dataset. Then, we adopt the episodic training procedure under 5-way 1-shot and 5-shot settings. In each episode, 16 unlabeled query images per class are used for the meta-training and meta-testing phases. We use SGD optimizer with a momentum of 0.9 and a weight decay of 5e-4. The outer-loop learning rate is initialized as 1e-3 on *mini*ImageNet datasets and 2e-4 on *tiered*ImageNet dataset. Our model is meta-trained for 100 epochs on all datasets. To ensure the stability of the evaluation results, we test 1,000 episodes and report the average performance with 95% confidence intervals. The inner-loop learning rate is initialized as 0.6. The number of inner-loop update steps is set to 25. The hyper-parameter $\tau$ is set as 1 for 1-shot setting, 0.2 for 5-shot setting in the inner loop, and 0.1 in the outer loop.

## B    ABLATION STUDY

**Influence of Inner-Loop Temperature.** To study the influence of inner-loop temperature hyper-parameter, we conduct experiments of FITE-RoBERTa with different inner-loop temperature values in our method. The rest of the settings are consistent with Section 5.1. Table 7 shows the results in 5-way 5-shot setting. We find that 0.2 is an appropriate inner-loop temperature value for this setting on *mini*ImageNet and *tiered*ImageNet datasets.

| Inner-Loop Temperature | *mini*ImageNet | *tiered*ImageNet |
|:---:|:---:|:---:|
| 1 | 77.56±0.50 | 78.14±0.63 |
| 0.7 | 80.51±0.46 | 81.70±0.57 |
| 0.5 | 82.42±0.43 | 83.74±0.54 |
| 0.3 | 83.49±0.41 | 86.33±0.49 |
| **0.2** | **83.90±0.40** | **87.67±0.45** |
| 0.1 | 82.11±0.41 | 86.24±0.48 |

Table 7: Ablation studies on the inner-loop temperature.

**Effect of the Number of Inner-Loop Update Steps.** To find a suitable number of inner-loop update steps, we keep the experimental setup in Section 5.1 and update the FITE-RoBERTa 15, 20, 25 and 30 steps in the inner loop respectively. Table 8 shows the results in 5-way 5-shot setting on *mini*ImageNet and *tiered*ImageNet. Following the results, we set the number of inner-loop update steps to 25 in our experiments.

| Number of Steps | 15 | 20 | **25** | 30 |
|:---:|:---:|:---:|:---:|:---:|
| *mini*ImageNet | 83.20±0.40 | 83.28±0.40 | **83.90±0.40** | 83.10±0.41 |
| *tiered*ImageNet | 86.14±0.49 | 86.02±0.33 | **87.67±0.45** | 86.61±0.48 |

Table 8: Ablation studies on the number of inner-loop update steps.

**Analyze of Learnable Prompt Template.** To evaluate the effect of the learnable prompt template, we adopt RoBERTa-base as the textual encoder and conduct experiments with fixed and learnable prompt templates. As shown in Table 9, learnable prompt template outperforms fixed prompt template by 1.97% on *mini*ImageNet and 1.26% on *tiered*ImageNet respectively.

**Analyze of Metric Module.** As is shown in Table 10, we explore replacing the bilinear form of the metric module in FILM-RoBERTa with other forms. We concatenated the visual and textual embeddings and used two MLPs to compute the similarity scores. The results show a significant decrease in performance on *mini*ImageNet and *tiered*ImageNet datasets, demonstrating the importance of the bilinear function. The metric module could be any function that computes the similarity between visual and textual embeddings. The bilinear function has many advantages, such as easy

| Learnable Prompt Template | *mini*ImageNet | *tiered*ImageNet |
|:---:|:---:|:---:|
| ✗ | 68.46±0.57 | 72.26±0.68 |
| ✓ | **70.43±0.61** | **73.52±0.71** |

Table 9: Ablation study on learnable prompt template. Reported are the accuracies on *mini*ImageNet and *tiered*ImageNet in a 5-way 1-shot setting. "✗" means that we fix the prompt template. "✓" means that we make the prompt template learnable in the outer loop.

| Metric Module | *mini*ImageNet | | *tiered*ImageNet | |
|:---:|:---:|:---:|:---:|:---:|
| | 1-shot | 5-shot | 1-shot | 5-shot |
| concatenation + MLP | 66.45±0.63 | 78.56±0.57 | 69.12±0.71 | 72.63±0.65 |
| **bilinear function** | **70.43±0.58** | **83.90±0.40** | **73.52±0.71** | **87.67±0.45** |

Table 10: Comparison between different metric modules.

implementation, few parameters, and little time consumption. To notice that, we adopt this simplest form in our experiments and achieve good performance.

## C   EXPERIMENTS ON META-DATASET

To further show the efficacy of the proposed method, we conduct experiments on several other datasets with semantic category names in the meta-dataset(Triantafillou et al., 2019). The results are presented in Table 11. Although with a weaker backbone (ResNet-12) than that used in meta-dataset (ResNet-18), our method outperforms all the baselines reported in meta-dataset. Besides, the performance on the meta-dataset also proves the strong cross-domain ability of our method as the model is only trained on the meta-training dataset of *mini*ImageNet.

| Method | Textures | VGG Flower | Quick Draw | Fungi | MSCOCO |
|:---:|:---:|:---:|:---:|:---:|:---:|
| ProtoNet | 66.56 | 85.27 | 48.96 | 39.71 | 41.00 |
| RelationNet | 52.97 | 68.76 | 43.30 | 30.55 | 29.15 |
| fo-Proto-MAML | 66.49 | 87.15 | 51.52 | 39.66 | 43.74 |
| FITE-CLIP (Ours) | **71.06±0.79** | **88.73±0.64** | **68.89±0.79** | **60.27±0.86** | **60.57±0.87** |

Table 11: Accuracies on meta-dataset in 5-way 5-shot setting.

## D   VISUALIZATION

**Visualization of $t$-SNE.** To qualitatively evaluate our method, we apply $t$-SNE (Laurens et al., 2008) to visualize the results, which represent the visual features and textual embeddings of five categories. We randomly sample 100 examples for each class in 5-way 1-shot setting on *mini*ImageNet dataset. As shown in Figure 3, the $t$-SNE visualization results indicate that our method can effectively align the visual embeddings (dots) and textual embeddings (stars) in the embedding space.

**Visualization of Grad-CAM.** In Figure 4, we use the Grad-CAM (Zhou et al., 2016) to visualize the pre-trained model and our method under a ResNet-12 feature extractor. It is observed that our method makes the model pay more attention to the discriminative part of the target object than the pre-trained model. For example, we find that for dog samples, the pre-trained model pays more attention to the background parts while our model focuses on the head part.

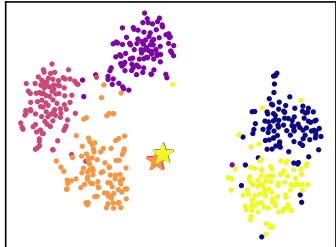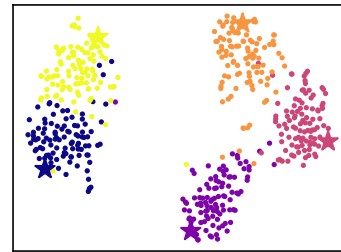

Figure 3: $t$-SNE visualization of the distribution of without the metric module (left) and using the metric module (right) with 5-way setting on *mini*ImageNet dataset. Dots and stars in different colors stand for visual embeddings and textual embeddings of different categories.

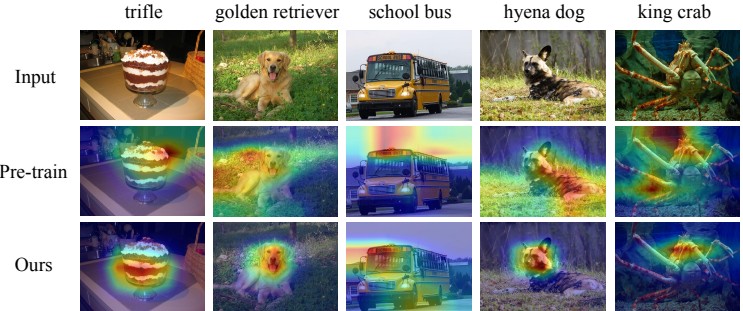

Figure 4: Grad-CAM visualization of *mini*ImageNet dataset.

# E    DISCUSSION ABOUT META-LEARNING PARADIGM

Meta-learning refers to a general training strategy that updates the model based on a series of "meta-tasks", encompassing various approaches, including "model-agnostic" meta-learning method (MAML) used in this paper. For MAML, one needs to determine which components are being updated in the inner loop and which components are being updated in the outer loop. Various possible solutions exist. Among these, we carefully design the metric module as a task-specific module and update other modules, such as the feature extractor, in the outer loop. Intuitively, feature extraction should be general for different downstream tasks. However, each concrete downstream task may have different focuses on feature channels. Therefore, the image-text alignment conducted by our metric module is designed to be task-specific and updated in the inner loop.

# F    DIFFERENCE FROM ZERO-SHOT LEARNING METHODS

Zero-shot learning methods(Xu et al., 2022) such as CLIP align image and text on the training dataset, in contrast, our few-shot learning method aims at quickly re-aligning image and text on few-shot datasets which may have totally different distributions from the meta-training dataset. Therefore, few-shot learning methods have distinct goals from zero-shot learning methods, and the performance of zero-shot learning methods may not translate well to few-shot learning scenarios, as they involve different core ideas.

Frozen(Tsimpoukelli et al., 2021) is the pioneer in using the off-the-shelf pretrained language model for both zero-shot and few-shot learning. There are several key differences between our method and Frozen. Firstly, Frozen encodes images into the word embedding space of a large pre-trained

language model and lets the language model generate captions for images, which solves discriminative problems in a generative manner. In contrast, our method introduces the concept of alignment, which is more suitable for classification tasks, leading to superior performance in few-shot classification tasks. Secondly, Frozen focuses on training a large multimodal model that can be generalized to various multimodal tasks in both zero-shot and few-shot settings. However, it does not thoroughly investigate about leveraging support examples to enable the model to quickly adapt to unseen datasets. In contrast, our method aims at quickly re-aligning image and text on unseen datasets with unseen categories. Specifically, we introduce a task-specific metric module and design a meta-learning framework, achieving strong generalization performance on meta-test datasets.

