# OpenReview forum: "Rethinking Semantic Few-Shot Image Classification"
_ICLR.cc/2024/Conference — Submitted to ICLR 2024_

### Official Review · Reviewer_72Kp · 2023-10-30

**Soundness:** 2 fair
**Presentation:** 2 fair
**Contribution:** 2 fair
**Rating:** 3
**Confidence:** 4

**Summary:**

This work tackles the problem of few-shot learning and proposes a method to adapt a model employing the MAML framework. Specifically, the model is updated to perform on a specific task inferred from the alignment loss between text and image embeddings. The metric module is introduced to extend the cosine similarity between two modalities. The experiments show that the proposed method can outperform SOTAs in few-shot learning on several benchmarks.

**Strengths:**

- This paper proves the efficacy of the proposed method by beating SOTAs on several benchmarks e.g., mini-Imagenet and CUB datasets.
- Ablation study is provided to understand the importance of the metric module.

**Weaknesses:**

- The novelty of this work is limited as the off-the-shelf text encoder has been proposed previously using a similar contrastive loss for few-shot learning. The idea has overlapping to the idea proposed in VS-alignment (Afham et al., 2022) with a marginal extension in the meta-learning technique with MAML.  Some discussion (head-to-head) on the proposed method and Afham et al. would be beneficial for the readers to spot the difference and novelty of the work.
- The citation to Meta learning paradigm in Page 3 is not precise. Vinyals et al., 2016  do not discuss about meta-learning but the work is more related to learn in a few data regime. MAML paper would be a more relevant citation in this part.
- The manuscript is not well written. Equation (3?) in Page 6 is not precisely correct as the gradient descent should be performed w.r.t. I, T, and M. Please check the expression after $\nabla$. Also, the equation in Page 6 has no number. Please fix this in the revised version.
- Regarding this sentence in Page 2: “Secondly, in contrast to vision-language pre-trained models where both visual and textual encoders are learnable to align embeddings, we utilize frozen public textual encoders. This leads to totally different structures of textual embedding spaces and thus makes the alignment between visual and textual features difficult,” One concern is that why we cannot use the image encoder from a pretrained image encoder (e.g., CLIP image encoder)? Are there any settings that do not allow this condition? If this was feasible, would this work still consider MAML with several gradient steps to obtain optimal performance?
- This work must consider one important work [1] in this direction.  This work is the pioneer in using the off-the-shelf pretrained language model for few-shot learning but there is no discussion and citation to this work.
- The experiments are quite limited as most of the comparisons only involve image data without text information. It would be better to provide some other datasets e.g., COCO. Some other experiments as in [1] might be considered to show the efficacy of the proposed method and comparison with [1] on different modes (e.g., frozen and not frozen).

[1] Tsimpoukelli et al., "Multimodal Few-Shot Learning with Frozen Language Models," Neurips, 2021.

**Questions:**

Please see the weaknesses, especially the question about the image encoder.

---

> ### Author Response · Authors · 2023-11-19
> **Response to Reviewer 72Kp (1/3)**
>
> **Q1.** The novelty of this work is limited as the off-the-shelf text encoder has been proposed previously using a similar contrastive loss for few-shot learning. The idea has overlapping to the idea proposed in VS-alignment (Afham et al., 2022) with a marginal extension in the meta-learning technique with MAML. Some discussion (head-to-head) on the proposed method and Afham et al. would be beneficial for the readers to spot the difference and novelty of the work.
>
> **A1.** Thank you for your valuable feedback. While both VS-alignment and our method utilize contrastive learning to align visual and textual features, they differ in several key aspects.
>
> Firstly, VS-alignment directly aligns the visual and textual features in a manner similar to CLIP. However, for few-shot tasks, **direct alignment can result in very poor performance**, as illustrated in Figure 1 of our paper. This is because the meta-test task and meta-training task may have a large distribution gap in few-shot learning, indicating that alignment may be no longer satisfied for the meta-test task. In our work, **we address this limitation by introducing a task-specific image-text metric module**, which can adaptively align the visual and textual features. As a result, the experimental results in Table 1 of our paper demonstrate **substantial performance improvement**, rising from 65.89% to 74.48% over VS-alignment.
>
> Secondly, our contribution extends beyond the design of a metric module for aligning text embeddings and image embeddings. More importantly, **we introduce a novel meta-learning framework for learning the parameters of our method**. In particular, we carefully design the metric module as a task-specific component, and make other modules such as the feature extractor updating in the outer loop.
>
> Thirdly,  **our work demonstrates the feasibility of aligning fixed text embeddings extracted from pre-trained language models with visual features**. As shown in Table 1 of our paper, it is noteworthy that our FITE-GloVe and FITE-RoBERTa exhibit a remarkable ability to achieve performances closely aligned with that of FITE-CLIP.
>
> These distinctions emphasize the uniqueness and advancements introduced in our approach compared to the VS-alignment method, providing a clearer understanding of the novelty in our work.
>
> **Q2.** The citation to Meta learning paradigm in Page 3 is not precise. Vinyals et al., 2016 do not discuss about meta-learning but the work is more related to learn in a few data regime. MAML paper would be a more relevant citation in this part.
>
> **A2.** Thank you for your feedback. The term "meta-learning paradigm" refers to a training strategy that updates the model based on a series of meta-tasks, encompassing various approaches, including but not limited to "model-agnostic" meta-learning methods such as MAML (Finn et al., 2017). Notably, MatchingNet (Vinyals et al., 2016) is recognized as one of the pioneering efforts that introduced the meta-learning paradigm into few-shot learning.
>
> However, we acknowledge your suggestion and have incorporated the MAML paper in the citation to the Meta-learning paradigm on Page 3. We believe this adjustment enhances the precision of our references and clarifies the connection between our work and relevant meta-learning methodologies.
>
> **Q3.** The manuscript is not well written. Equation (3?) in Page 6 is not precisely correct as the gradient descent should be performed w.r.t. I, T, and M. Please check the expression after ∇. Also, the equation in Page 6 has no number. Please fix this in the revised version.
>
> **A3.** Thank you for your feedback. We would like to clarify that only the metric module (M) is designed to be updated in the inner loop, while the image and text encoders (I and T) remain unchanged in the inner loop. Therefore, the gradient descent should not be performed with respect to I and T. As such, the equation in Page 6 is accurate as stated.
>
> Regarding the equation numbering, we follow Occam's Rule (http://www.ams.org/notices/199508/hwang.pdf): Number only those equations that are referred back to. We appreciate your understanding and will ensure that the necessary corrections are made in the revised version.

---

> ### Author Response · Authors · 2023-11-19
> **Response to Reviewer 72Kp (2/3)**
>
> **Q4.** Regarding this sentence in Page 2: “Secondly, in contrast to vision-language pre-trained models where both visual and textual encoders are learnable to align embeddings, we utilize frozen public textual encoders. This leads to totally different structures of textual embedding spaces and thus makes the alignment between visual and textual features difficult,” One concern is that why we cannot use the image encoder from a pretrained image encoder (e.g., CLIP image encoder)? Are there any settings that do not allow this condition? If this was feasible, would this work still consider MAML with several gradient steps to obtain optimal performance?
>
> **A4.** Thank you for your question. We would like to clarify that our work follows the standard semantic few-shot classification setting as established in [r1][r2][r3], wherein only pre-trained text encoders are available, and pre-trained image encoders (e.g., CLIP image encoder) are not accessible. Notably, our method can achieve state-of-the-art (SOTA) performance in this standard setting.
>
> If a pre-trained image encoder is allowed to access, the performance will surpass that of the aforementioned setting, making it unfair for direct comparison with previous works. However, if such access were feasible, employing MAML would be still considered to achieve better performance. This is because pre-trained CLIP focuses on aligning image and text on the training dataset, in contrast, our few-shot learning method aims at quickly re-aligning image and text on unseen datasets with unseen categories, which may have totally different distributions from the training dataset. MAML is specifically designed to address such scenarios, making it a valuable consideration for improving performance in this context.
>
> [r1] AM3: Adaptive cross-modal few-shot learning.
>
> [r2] LPE: Semantic guided latent parts embedding for few-shot learning.
>
> [r3] SP: Semantic prompt for few-shot image recognition.
>
> **Q5.** This work must consider one important work [1] in this direction. This work is the pioneer in using the off-the-shelf pretrained language model for few-shot learning but there is no discussion and citation to this work.
>
> [1] Tsimpoukelli et al., "Multimodal Few-Shot Learning with Frozen Language Models," Neurips, 2021.
>
> **A5.** Thank you for reminding us of this related paper. While both Frozen [1] and our method use the frozen pre-trained language model and train the visual encoder from scratch for the few-shot learning task, there are several key differences between them.
>
> Firstly, Frozen encodes images into the word embedding space of a large pre-trained language model and lets the language model generate captions for images, **which solves discriminative problems in a generative manner**. In contrast, **our method introduces the concept of alignment, which is more suitable for classification tasks**, leading to superior performance in few-shot classification tasks.
>
> Secondly, Frozen focuses on training a large multimodal model that can be generalized to various multimodal tasks in both zero-shot and few-shot settings. However, **it does not thoroughly investigate about leveraging support examples to enable the model to quickly adapt to unseen datasets**. In contrast, **our method aims at quickly re-aligning image and text on unseen datasets with unseen categories**. Specifically, we introduce a task-specific metric module and design a meta-learning framework, achieving strong generalization performance on meta-test datasets.
>
> The specific performance comparisons related to the above two points are discussed in **A6**. We have cited this paper and added the discussion about this work in the revised paper.

---

> ### Author Response · Authors · 2023-11-19
> **Response to Reviewer 72Kp (3/3)**
>
> **Q6.** The experiments are quite limited as most of the comparisons only involve image data without text information. It would be better to provide some other datasets e.g., COCO. Some other experiments as in [1] might be considered to show the efficacy of the proposed method and comparison with [1] on different modes (e.g., frozen and not frozen).
>
> **A6.** Thanks for your suggestion.
>
> Firstly, we have also conducted experiments on several other datasets with semantic category names in the meta-dataset [r4] (including COCO), and the results are presented in the table below.
> |  | Textures | VGG Flower | Quick Draw | Fungi | MSCOCO |
> | --- | --- | --- | --- | --- | --- |
> | ProtoNet | 66.56 | 85.27 | 48.96 | 39.71 | 41.00 |
> | RelationNet | 52.97 | 68.76 | 43.30 | 30.55 | 29.15 |
> | fo-Proto-MAML | 66.49 | 87.15 | 51.52 | 39.66 | 43.74 |
> | **FITE-CLIP** | **71.06±0.79** | **88.73±0.64** | **68.89±0.79** | **60.27±0.86** | **60.57±0.87** |
>
> Although with a weaker backbone (ResNet-12) than that used in meta-dataset (ResNet-18), our method outperforms all the baselines reported in meta-dataset. Besides, **the performance on the meta-dataset also proves the strong cross-domain ability of our method** as the model is only trained on the meta-training dataset of miniImageNet. We have added this experiment to the revised version of our paper.
>
> Secondly, our method exhibits stronger performance in few-shot classification scenarios. On the one hand, although Frozen uses a stronger backbone (e.g., NF-ResNet-50) and trains it on a large-scale multimodal dataset (e.g., Conceptual Captions), its performance is not significantly above chance (20.2% for 1-shot and 21.3% for 5-shot) on miniImagenet in the 5-way setting. In contrast, our FITE-CLIP achieves accuracies of 74.48% and 84.45% on miniImagenet in 5-way 1-shot and 5-way 5-shot settings respectively. On the other hand, as the number of examples increases, the performance improvement of our method (an increase of 9.97% from 1-shot to 5-shot) is more significant than Frozen (an increase of 1.1% from 1-shot to 5-shot). From the performance comparison, it can be observed that our method is more suitable for few-shot classification scenarios.
>
> [r4] Meta-Dataset: A Dataset of Datasets for Learning to Learn from Few Examples

---

> ### Author Response · Authors · 2023-11-20
> **We would be grateful if you could take a look at the response**
>
> Dear Reviewer 72Kp:
>
> We sincerely appreciate your valuable time devoted to reviewing our manuscript. We would like to gently remind you of the **approaching deadline for the discussion phase**. We have diligently addressed the issues you raised in your feedback, providing detailed explanations. For instance, we have explained the difference between VS-alignment and our method, highlighting the novelty of our method. Moreover, we have made a comparison between Frozen and our method and exhibited the superior performance of our method in few-shot classification scenarios. Besides, we have conducted experiments on meta-dataset (including COCO) to further show the efficacy of our method. Would you kindly take a moment to look at it?
>
> We are very enthusiastic about engaging in more in-depth discussions with you.

---

> ### Author Response · Authors · 2023-11-22
>
> Dear Reviewer 72Kp,
>
> As the discussion period approaches its conclusion, **we want to ensure that we have thoroughly addressed all your concerns and that our revised paper fully meets the standards of ICLR**. We would highly value any additional feedback you may provide.
>
> Thank you sincerely for your time and consideration.
>
> Best regards,
> The Authors

---

> ### Comment · Reviewer_72Kp · 2023-11-22
> **Thanks for the Response**
>
> Thanks for the response, I have read the rebuttal and the edited version.
> Some answers have only answered my concerns partially:
> - Q1. I still think the novelty is quite marginal as several works cited in the paper propose quite a similar approach.
> - Q2. As explained in the initial comment, few-shot learning has a different meaning compared to meta-learning. Though Matching Networks has proposed a few-shot learning problem, it is not a meta-learning problem. Hope this work could correct this misunderstanding.
> - Q3. In Sec 4.4, Equation (supposed to be 3) is not fixed in the edited version, it remains no number. I still feel this equation is not correct. I now can understand the gradient should be performed only on one component but couldn't understand from the description in the text. Please look at the meaning in the text: "In the inner loop, we adapt the model to the current
> task $T_j$ by updating $\theta_M^{'}$ with a number of gradient descent steps on the support set while keeping $\theta_I$ ,
>  $\theta_T$ and  $\theta_M$ fixed". This sentence and the equation are quite confusing. Which parameters are updated in the inner loop and outer loop?
> - Q4. Understood about the setting, thanks for explaining this. However, what does it stop this work to train an image encoder based on the base classes using a pretrained text encoder to encode text information? In this way, I believe this work would not violate the training and test settings in the mentioned works.
> - Q5. I wish this can be incorporated in the ablation study to verify the frozen LLM or text encoder capabilities for few-shot learning.
>
> In general, this work still requires some ideas refinement and more informative experimental results. Also, it needs to dissect each component and different configurations impacting the final results rather than only providing the final results and comparison to baselines.

---

### Official Review · Reviewer_rhJx · 2023-10-31

**Soundness:** 2 fair
**Presentation:** 2 fair
**Contribution:** 2 fair
**Rating:** 3
**Confidence:** 5

**Summary:**

This paper addresses few-shot image classification which is tasked to learn a classifier on new classes with only a few samples. This paper proposes to leverage class-level text-embeddings and introduces a metric module to align text and image embeddings. The text-embeddings in the experiments include language models, word embedding and CLIP text embeddings. The method follows the training and optimization framework of MAML, a popular meta-learning method. The experiments are conducted on the widely used few-shot learning benchmarks including miniImageNet, tieredImageNet and CUB. The results show better results than the baselines.

**Strengths:**

-The paper is written well and easy to understand.

-The proposed method is simple. It is intuitively a good idea to leverage the semantic information for few-shot image classification.

**Weaknesses:**

-Overclaim the technical novelty. The framework of learning image and text alignment with a bilinear function has been extensively studied in zero-shot learning.  Some zero-shot learning methods (e.g., Xu et al.) even show that the framework generalizes well to few-shot learning setting. Although this paper adopts a different few-shot learning setting with episodic evaluation protocol, in principle, the core method is rather similar. In my view, this work seems to be a trivial combination of MAML and previous zero-shot learning methods, which can not be counted as a significant technical contribution. This paper seems to ignore this point and fails to discuss how it improves the zero-shot learning methods.

[A] Xu et al., Attribute Prototype Network for Any-Shot Learning. IJCV 2022.

-Lack of insights. Although the proposed method achieves SOTA on some dataset, it mainly relies on the CLIP text embeddings, which is somewhat expected because it is known that CLIP embeddings can achieve impressive image classification results. The results using word embeddings and language embeddings are actually worse than some baseline methods. There are not much insights except the comparison between three different text embeddings.

**Questions:**

In Sec. 4.1, it says "Besides, we make the prompt templates learnable to avoid time-consuming prompt engineering following." But I did not find any technical detail about this part.

---

> ### Author Response · Authors · 2023-11-19
> **Response to Reviewer rhJx (1/2)**
>
> **Q1.** Overclaim the technical novelty. The framework of learning image and text alignment with a bilinear function has been extensively studied in zero-shot learning.
>
> Some zero-shot learning methods (e.g., Xu et al.) even show that the framework generalizes well to few-shot learning setting. Although this paper adopts a different few-shot learning setting with episodic evaluation protocol, in principle, the core method is rather similar.
>
> In my view, this work seems to be a trivial combination of MAML and previous zero-shot learning methods, which can not be counted as a significant technical contribution. This paper seems to ignore this point and fails to discuss how it improves the zero-shot learning methods.
>
> [A] Xu et al., Attribute Prototype Network for Any-Shot Learning. IJCV 2022.
>
> **A1.** Thanks for your comments.
>
> First of all, we would like to clarify that the metric module of our method is designed as a general learnable function for measuring the alignment between image and text, which is **not limited to the bilinear function**. Our method is suitable for other function forms beyond the bilinear function.
>
> Second, zero-shot learning methods such as CLIP align image and text on the training dataset, in contrast, our few-shot learning method aims at **quickly re-aligning image and text on few-shot datasets which may have totally different distributions from the meta-training dataset**. Therefore, **few-shot learning methods have distinct goals from zero-shot learning methods**, and the performance of zero-shot learning methods may not translate well to few-shot learning scenarios, as they involve different core ideas.
>
> Thirdly, even when considering the combination of MAML and zero-shot learning methods, there exist various possible solutions, and our approach is not a trivial combination. Among these possible solutions, **we carefully design the metric module as a task-specific module and make other modules such as the feature extractor updated in the outer loop**. Intuitively, feature extraction should be general for different downstream tasks. However, each concrete downstream task may have different focuses on feature channels. Therefore, the image-text alignment conducted by our metric module should be task-specific and updated in the inner loop.
>
> We have cited this paper and added the discussion about zero-shot learning methods in the revised paper.
>
> **Q2.** Lack of insights. Although the proposed method achieves SOTA on some dataset, it mainly relies on the CLIP text embeddings, which is somewhat expected because it is known that CLIP embeddings can achieve impressive image classification results. There are not much insights except the comparison between three different text embeddings.
>
> **A2.** Firstly, it is expected that CLIP text embeddings achieve better performance than word embedding and language embeddings, as CLIP text embeddings are trained with contrastive learning on large-scale image-text pairs data. However, we would like to clarify that **our work provides an insight that textual embeddings trained without multimodal data (e.g., GloVe, RoBERTa) can also be aligned with visual features**. In particular, how to align the text embeddings extracted from the language model pre-trained on text corpora and image embeddings is a challenging problem. As shown in Figure 1 of our paper, direct alignment results in bad performance. To address the problem, we introduce a metric module and design a meta-learning framework. By doing so, our FITE-GloVe and FITE-RoBERTa can achieve performance close to FITE-CLIP.
>
> Secondly, while CLIP embeddings can indeed achieve impressive image classification results, we would like to highlight that **our FITE-CLIP achieves SOTA performance on various datasets compared to other baselines that also adopt pre-trained CLIP embeddings** (e.g., SP-CLIP). This underscores the effectiveness of our approach where the metric module and MAML framework are introduced.
>
> Thirdly, it's important to note that **semantic few-shot learning SOTA methods typically focus on feature fusion between visual and textual embeddings** (e.g., SP-CLIP). In contrast, **our method introduces the concept of alignment**, leading to superior performance. This suggests that alignment offers a more promising approach compared to feature fusion in semantic few-shot learning.
>
> In conclusion, our work not only leverages the expected advantages of CLIP embeddings but also offers novel insights into the alignment between visual and text embeddings in the few-shot learning problem.

---

> > ### Comment · Reviewer_rhJx · 2023-11-22
> >
> > Thanks for the response. However, my concerns are only partially addressed by the response.
> >
> > 1. The rebuttal fails to discuss the technical differences between the alignment module in this paper and the classical zero-shot learning approaches [Akata et al., CVPR'15]. The main argument in the rebuttal is that zero-shot and few-shot learning have different goals. However, adapting zero-shot learning methods to few-shot learning does not have sufficient technical contributions for ICLR in my point of view. Moreover, the response says "the performance of zero-shot learning methods may not translate well to few-shot learning scenarios". However, this claim is not supported by any evidence.
> >
> > 2. Regarding the missing insights. The rebuttal says that "our work provides an insight that textual embeddings trained without multimodal data (e.g., GloVe, RoBERTa) can also be aligned with visual features". Unfortunately, this is not something new as this has been known for a long time in zero-shot learning field, for example [Akata et al., CVPR'15].
> >
> > Akata et al., Evaluation of Output Embeddings for Fine-Grained Image Classification. CVPR'15

---

> ### Author Response · Authors · 2023-11-19
> **Response to Reviewer rhJx (2/2)**
>
> **Q3.** The results using word embeddings and language embeddings are actually worse than some baseline methods.
>
> **A3.** Thank you for your comments. Firstly, we would like to emphasize that our baseline, SP, selects ViT as the backbone, which is necessary for their feature fusion framework, and requires additional augmentation (i.e., RepeatAug), while our method FITE utilizes a standard ResNet-12 backbone and does not employ data augmentation. Despite this, our FITE-GloVe and FITE-RoBERTa still achieve comparable performance with SP-GloVe and SP-SBERT. Additionally, we would like to draw the reviewer's attention to the fact that our FITE-CLIP outperforms all the baselines using CLIP including SP-CLIP.
>
> Since the backbone of our method is flexible, we also conducted experiments with ViT and Swin backbones for FITE-CLIP, and the results are presented in the table below.
> | Backbone  | miniImageNet |  | tieredImageNet |  |
> | --- | --- | --- | --- | --- |
> |  | 1 shot | 5 shot | 1 shot | 5 shot |
> | ResNet-12 | 74.48±0.55 | 84.45±0.40 | 79.20±0.43 |  88.85±0.13 |
> | ViT-Small | 74.82±0.76 | 86.19±0.54 | 81.38±0.88 | 89.53±0.61 |
> | Swin-Tiny | **76.49±0.75** | **87.33±0.54** | **81.98±0.80** | **90.48±0.54** |
>
> From the table, we observe that **the replacement of the backbone from ResNet-12 with ViT-Small and Swin-Tiny yields significant performance improvements**. The results also suggest that **our approach can adapt well to different types of backbones without the need for adjustments**. We have incorporated this ablation study into the revised version of our paper.
>
> **Q4.** In Sec. 4.1, it says "Besides, we make the prompt templates learnable to avoid time-consuming prompt engineering following." But I did not find any technical detail about this part.
>
> **A4.** Sorry for the confusion. Specifically, the word embedding of each token in the prompt template is replaced by a learnable vector of the same dimension. These learnable vectors are initialized with original word embeddings and updated in the outer loop together with other parameters. We have added this detail to the revised version of our paper.

---

> ### Author Response · Authors · 2023-11-20
> **We would be grateful if you could take a look at the response**
>
> Dear Reviewer rhJx:
>
> We sincerely appreciate your valuable time devoted to reviewing our manuscript. We would like to gently remind you of the **approaching deadline for the discussion phase**. We have diligently addressed the issues you raised in your feedback, providing detailed explanations. For instance, we have explained the difference between the core idea of zero-shot learning works and that of our method. Moreover, we have clarified the insights provided by our work. Besides, we have added the technical details about the learnable prompt template. Would you kindly take a moment to look at it?
>
> We are very enthusiastic about engaging in more in-depth discussions with you.

---

> ### Author Response · Authors · 2023-11-22
>
> Dear Reviewer rhJx,
>
> As the discussion period approaches its conclusion, **we want to ensure that we have thoroughly addressed all your concerns and that our revised paper fully meets the standards of ICLR**. We would highly value any additional feedback you may provide.
>
> Thank you sincerely for your time and consideration.
>
> Best regards,
> The Authors

---

### Official Review · Reviewer_Pbba · 2023-11-01

**Soundness:** 2 fair
**Presentation:** 3 good
**Contribution:** 2 fair
**Rating:** 3
**Confidence:** 4

**Summary:**

This paper is about few shot image classification. It aims at improving it by exploiting semantic information that takes the form of textual embedding obtained using existing textual encoders. As this textual information is not directly aligned with the visual one, the authors propose to align them by metric learning and in particular contrastive learning widely used in self-supervised representation learning. This training procedure follows the metal-learning approach and in particular MAML. Experimental validation and comparison to the state-of-the-art are done on two few shot benchmarks (miniImagenet and tieredImageNet) and the CUB-200-2011 dataset for fine-grained recognition. Their approaches improve the performances on these benchmarks. An ablation study on the metric learning part is provided.

**Strengths:**

+ The paper tackles an important issue in few-shot learning, i.e. improving existing approaches by injecting additional information and in particular semantic textual information. Indeed, this line of work has been largely studied recently and has shown promising results.
+ The paper is well-written with clear objectives and motivations.
+ The experimental part implies a comparison with various inductive state-of-the-art approaches and the obtained results show an improvment in terms of performance on different 5 ways x shot settings.
+ The paper also contains a small ablation study on the metric learning part.

**Weaknesses:**

+ A first concern is about the experimental study which is incomplete from my point of view on several aspects :
  +  First, it is usual to study the dependence on the visual backbone. Only Resnets-12  is used in the paper but other backbones such as visual transformers backbone or recent foundation models could be included.
  + The influence of the prompting strategy could also be experimented with in more detail. The appendix provides an analysis of the learnable prompt template but it could have been interesting to correlate this prompting strategy to a more formalized definition of the type of semantic information that should be carried.
  + Some technical details are missing. In particular, it is well known that contrastive learning is highly dependent on the negative sampling strategy but also on the size of the batch. This information is missing in the paper.
 + Some benchmarks have been provided in the Few Shot community to better take into account the semantics. See for instance [meta-dataset](https://github.com/google-research/meta-dataset) or the work presented [here](https://arxiv.org/abs/2205.05155). What is the behavior of the approach on these benchmarks?

+ Contrastive learning has been studied in the context of multimodal data. The positioning to these works could be interesting. See for instance [this paper](https://openaccess.thecvf.com/content/CVPR2021/papers/Yuan_Multimodal_Contrastive_Training_for_Visual_Representation_Learning_CVPR_2021_paper.pdf).

+ Another concern is about the novelty of the proposed approach compared to (Chen et al, 2023). Compared to this work, the authors propose to add the metric module but this latter is also not new. I would appreciate it if the authors argued more on the novelty of this part, maybe in relation to the few shot settings and the way to tackle support and query sets.

**Questions:**

+ Recent works have shown the prevalence of transductive few-shot learning in image classification with this transductive setting that outperforms inductive approaches. How is the proposed approach compared to transductive sota approaches? How adapting the proposed approach to this scheme?
+ It is possible to better formalize the notion of semantic information? What kind of information should be added?
+ The meta-learning paradigm has been discussed a lot in the few-shot community. See for instance [this paper](https://arxiv.org/abs/2003.11539). A discussion on this point should be added in the paper.

---

> ### Author Response · Authors · 2023-11-19
> **Response to Reviewer Pbba (1/3)**
>
> **Q1.** First, it is usual to study the dependence on the visual backbone. Only Resnets-12 is used in the paper but other backbones such as visual transformers backbone or recent foundation models could be included.
>
> **A1.** Thanks for your suggestion. We only use ResNet-12 in our paper as it is the standard backbone used in the few-shot classification setting. For example, many highly-cited few-shot papers also only use ResNet-12 in their experiments, such as
> - AM3: Adaptive Cross-Modal Few-shot Learning
> - TRAML: Boosting Few-Shot Learning With Adaptive Margin Loss
> - RENet: Relational Embedding for Few-Shot Classification
> - DeepEMD: Few-Shot Image Classification with Differentiable Earth Mover’s Distance and Structured Classifiers
>
> Moreover, in response to your suggestion, we also conduct experiments with ViT-Small and Swin-Tiny, and the results are presented in the table below.
> | Backbone | miniImageNet |  | tieredImageNet |  |
> | --- | --- | --- | --- | --- |
> |  | 1 shot | 5 shot | 1 shot | 5 shot |
> | ResNet-12 | 74.48±0.55 | 84.45±0.40 | 79.20±0.43 |  88.85±0.13 |
> | ViT-Small | 74.82±0.76 | 86.19±0.54 | 81.38±0.88 | 89.53±0.61 |
> | Swin-Tiny | **76.49±0.75** | **87.33±0.54** | **81.98±0.80** | **90.48±0.54** |
>
> From the table, we observe that **the replacement of the backbone from ResNet-12 with more powerful ViT-Small and Swin-Tiny yields significant performance improvements**. The results also suggest that **our approach can adapt well to different types of backbones without the need for adjustments**. We have incorporated this ablation study into the revised version of our paper.
>
> **Q2.** The influence of the prompting strategy could also be experimented with in more detail. The appendix provides an analysis of the learnable prompt template but it could have been interesting to correlate this prompting strategy to a more formalized definition of the type of semantic information that should be carried.
>
> **A2.** Sorry for the confusion. In our paper, "semantic information" and "textual embedding" are used interchangeably without differentiation, following the terminology used in previous semantic few-shot learning works (e.g. SP [r1]). In particular, we design the prompting strategy and use a learnable prompt template, obtaining textual embeddings that are better for alignment in specific tasks.
>
> [r1] Semantic prompt for few-shot image recognition
>
> **Q3.** Some technical details are missing. In particular, it is well known that contrastive learning is highly dependent on the negative sampling strategy but also on the size of the batch. This information is missing in the paper.
>
> **A3.** Thank you for your feedback. We would like to clarify that **some concepts in contrastive learning will have slight variations when applied to the few-shot setting**. Specifically, the few-shot learning problem is formulated as N-way K-shot classification (see the problem definition section). Taking 5-way 1-shot as an example, a fixed number of support samples (5) are loaded in each episode. The contrastive loss should be calculated among all these samples rather than selecting some hard negative samples. Consequently, the negative sample size is fixed and equal to 5-1=4. Therefore, due to the few-shot learning setting, we can not change the negative sampling strategy or the negative sample size.
>
> **Q4.** Some benchmarks have been provided in the Few Shot community to better take into account the semantics. See for instance meta-dataset or the work presented here. What is the behavior of the approach on these benchmarks?
>
> **A4.** We choose widely used few-shot datasets (e.g., miniImageNet, tieredImageNet, and CUB) as benchmarks following the baselines in our paper. Following your suggestion, we have also conducted experiments on several other datasets with semantic category names in the meta-dataset. The results are presented in the table below.
> |  | Textures | VGG Flower | Quick Draw | Fungi | MSCOCO |
> | --- | --- | --- | --- | --- | --- |
> | ProtoNet | 66.56 | 85.27 | 48.96 | 39.71 | 41.00 |
> | RelationNet | 52.97 | 68.76 | 43.30 | 30.55 | 29.15 |
> | fo-Proto-MAML | 66.49 | 87.15 | 51.52 | 39.66 | 43.74 |
> | **FITE-CLIP** | **71.06±0.79** | **88.73±0.64** | **68.89±0.79** | **60.27±0.86** | **60.57±0.87** |
>
> Although with a weaker backbone (ResNet-12) than that used in meta-dataset (ResNet-18), our method outperforms all the baselines reported in meta-dataset. Besides, **the performance on the meta-dataset also proves the strong cross-domain ability of our method** as the model is only trained on the meta-training dataset of miniImageNet. We have added this experiment to the revised version of our paper.

---

> ### Author Response · Authors · 2023-11-19
> **Response to Reviewer Pbba (2/3)**
>
> **Q5.** Contrastive learning has been studied in the context of multimodal data. The positioning to these works could be interesting. See for instance this paper.
>
> **A5.** Different from previous studies (e.g., [r2]) about contrastive learning on multimodal data, our paper focuses on applying multimodal contrastive learning to few-shot scenarios. Previous studies (e.g., [r2]) aim at training the model on large-scale multimodal datasets (e.g., COCO and Stock) to obtain high-quality multimodal features for downstream tasks. In contrast, **only small-scale few-shot datasets and fixed pre-trained language models are available in the semantic few-shot setting, which makes it more difficult to align visual and textual features, especially with language models pre-trained without multimodal data**. To address the problem, we introduce a metric module and design a meta-learning framework, achieving alignment quickly on meta-test datasets with three types of language models. We have added the discussion about this series of works in the revised version of our paper.
>
> [r2] Multimodal Contrastive Training for Visual Representation Learning
>
> **Q6.** Another concern is about the novelty of the proposed approach compared to (Chen et al, 2023). Compared to this work, the authors propose to add the metric module but this latter is also not new. I would appreciate it if the authors argued more on the novelty of this part, maybe in relation to the few shot settings and the way to tackle support and query sets.
>
> **A6.** Thank you for your valuable feedback. While both SP (Chen et al, 2023) and our method utilize semantic information to improve few-shot classification, they differ in several key aspects.
>
> Firstly, we would like to clarify that **SP and our method perform semantic few-shot classification from two different lines of thought**. Semantic few-shot learning SOTA methods (including SP) typically focus on **feature fusion** between visual and textual embeddings. In contrast, our method introduces the concept of **alignment**, leading to superior performance. This suggests that alignment offers a more promising approach compared to feature fusion in semantic few-shot learning.
>
> Secondly, our contribution is not only the design of a metric module to align the text embeddings and image embeddings, but more importantly, **the design of a meta-learning framework for learning the parameters of the metric module in the few-shot scenario**. Our task-specific metric module leverages the support set, enhancing the alignment of text embeddings and image embeddings on the query set and leading to better performance on meta-test tasks.
>
> **Q7.** Recent works have shown the prevalence of transductive few-shot learning in image classification with this transductive setting that outperforms inductive approaches. How is the proposed approach compared to transductive sota approaches? How adapting the proposed approach to this scheme?
>
> **A7.** Thanks for your question. We would like to clarify that **inductive few-shot learning (FSL) and transductive FSL are two different branches of few-shot learning**. The former predicts query samples one by one while the latter predicts query samples as a whole. The transductive setting leverages the statistics of the unlabeled query set and thus generally outperforms inductive inference. Consequently, inductive methods and transductive methods are typically compared independently in the literature. Therefore, it is unfair to compare our approach for inductive FSL with transductive FSL methods. As the methods of these two branches employ quite different techniques, inductive approaches cannot be directly adapted to the transductive FSL scheme. However, we observe that MAML can be adapted to the transductive FSL scheme [r3]. Thus, the extension of our work to transductive FSL is considered a potential direction for future work, and we acknowledge this as an avenue for further exploration.
>
> [r3] Leveraging the Feature Distribution in Transfer-based Few-Shot Learning
>
> **Q8.** It is possible to better formalize the notion of semantic information? What kind of information should be added?
>
> **A8.** As this question is similar to **Q2**, please see **A2** for the explanation of the semantic information.

---

> ### Author Response · Authors · 2023-11-19
> **Response to Reviewer Pbba (3/3)**
>
> **Q9.** The meta-learning paradigm has been discussed a lot in the few-shot community. See for instance this paper. A discussion on this point should be added in the paper.
>
> **A9.** Thanks for your suggestion. Meta-learning refers to a general training strategy that updates the model based on a series of "meta-tasks", encompassing various approaches, including "model-agnostic" meta-learning method (MAML) used in this paper. For MAML, one needs to determine which components are being updated in the inner loop and which components are being updated in the outer loop. Various possible solutions exist. Among these, we carefully design the metric module as a task-specific module and update other modules, such as the feature extractor, in the outer loop. Intuitively, feature extraction should be general for different downstream tasks. However, each concrete downstream task may have different focuses on feature channels. Therefore, the image-text alignment conducted by our metric module is designed to be task-specific and updated in the inner loop.
>
> We have added the discussion about the meta-learning paradigm in the revised paper.

---

> ### Author Response · Authors · 2023-11-20
> **We would be grateful if you could take a look at the response**
>
> Dear Reviewer Pbba:
>
> We sincerely appreciate your valuable time devoted to reviewing our manuscript. We would like to gently remind you of the **approaching deadline for the discussion phase**. We have diligently addressed the issues you raised in your feedback, providing detailed explanations. For instance, we have conducted experiments to study the dependence on the visual backbone using ViT-Small and Swin-Tiny. Moreover, we have conducted experiments on meta-dataset to further show the efficacy of our method. Besides, we have explained the difference between SP and our method, highlighting the novelty of our method. Would you kindly take a moment to look at it?
>
> We are very enthusiastic about engaging in more in-depth discussions with you.

---

> ### Author Response · Authors · 2023-11-22
>
> Dear Reviewer Pbba,
>
> As the discussion period approaches its conclusion, **we want to ensure that we have thoroughly addressed all your concerns and that our revised paper fully meets the standards of ICLR**. We would highly value any additional feedback you may provide.
>
> Thank you sincerely for your time and consideration.
>
> Best regards,
> The Authors

---

> > ### Comment · Reviewer_Pbba · 2023-11-22
> > **Response to the rebuttal**
> >
> > Thanks to the authors for their detailed responses. They thoroughly addressed several concerns in the paper. It clearly improves the paper but I still think that the technical novelty is not sufficient for ICLR avenue.

---

### Official Review · Reviewer_sXdY · 2023-11-04

**Soundness:** 3 good
**Presentation:** 3 good
**Contribution:** 2 fair
**Rating:** 5
**Confidence:** 4

**Summary:**

This paper proposes a novel few-shot learning framework for image classification that leverages semantic information extracted by a public textual encoder based on contrastive learning. The proposed approach addresses the challenge of alignment between visual features and textual embeddings obtained from public textual encoders and introduces a metric module to generalize the similarity measure. The metric module is designed to be adaptive to different few-shot tasks for better transferability, and MAML is adopted to train the model via bi-level optimization. The paper demonstrates the effectiveness of the proposed method through extensive experiments on multiple benchmarks with different domains. The main contributions of the paper are the proposed few-shot learning framework, the carefully designed textual branch of the framework, the metric module for generalizing the similarity measure, and the demonstration of the effectiveness of the proposed method through extensive experiments.

**Strengths:**

+ The paper has a clear and well-organized presentation of the proposed method, and the authors provide comprehensive experiments.

+ The proposed method aims to bridge the gap between visual and textual modalities, which is a good direction towards few-shot learning.

+ The visualization (sec. 5.4) looks good.

**Weaknesses:**

- There is a lack of comparison with other state-of-the-art approaches, such as TRIDENT [1], BAVARDAGE [2], PEMnE-BMS [3], etc, though these methods may not rely on pretrained vision-language models. Besides, the authors miss some references about referring to category names for few-shot learning [4].

- In the bi-level optimization, the metric module is updated in the inner loop and then all parameters are updated later in the outer loop. Could the authors provide any reasons or motivation for this training strategy?

- This paper aims to minimize the gap between the visual and textual modalities by introducing the metric module. It adds degrees of freedom so that there are some learnable parameters for bridging the gap. However, vision-language models are capable of zero-shot inference, therefore the way initializing the "bridge" between two modalities is crucial to the performance. Have the authors considered leveraging category name embedding for initialization [4]? By doing so, two modalities will be aligned automatically.



[1] Transductive Decoupled Variational Inference for Few-Shot Classification
[2] Adaptive Dimension Reduction and Variational Inference for Transductive Few-Shot Classification
[3] Squeezing Backbone Feature Distributions to the Max for Efficient Few-Shot Learning
[4] Exploiting Category Names for Few-Shot Classification with Vision-Language Models

**Questions:**

I expect to hear back from the authors regarding the below questions and concerns.

1. Comparison regarding other state-of-the-art few-shot approaches with established results on CUB or Mini-Imagenet.
2. Explanation of the motivation or insights of the training strategy, especially why only the metric module is updated in the inner loop and all parameters are updated in the outer loop.
3. Discussion about the category name initialization and possible comparison or further experiments by adding that strategy to existing approach.

---

> ### Author Response · Authors · 2023-11-19
> **Response to Reviewer sXdY (1/2)**
>
> **Q1.** There is a lack of comparison with other state-of-the-art approaches, such as TRIDENT [1], BAVARDAGE [2], PEMnE-BMS [3], etc, though these methods may not rely on pretrained vision-language models. Besides, the authors miss some references about referring to category names for few-shot learning [4].
>
> [1] Transductive Decoupled Variational Inference for Few-Shot Classification
>
> [2] Adaptive Dimension Reduction and Variational Inference for Transductive Few-Shot Classification
>
> [3] Squeezing Backbone Feature Distributions to the Max for Efficient Few-Shot Learning
>
> [4] Exploiting Category Names for Few-Shot Classification with Vision-Language Models
>
> **A1.** Thanks for your feedback.
>
> Firstly, we would like to clarify that **inductive few-shot learning (FSL) and transductive FSL are two different branches of few-shot learning**. The former predicts query samples one by one while the latter predicts query samples as a whole. The transductive setting leverages the statistics of the unlabeled query set and thus generally outperforms inductive inference. Consequently, inductive methods and transductive methods are typically compared independently in the literature. Therefore, **it is unfair to compare our approach for inductive FSL with transductive FSL methods** such as [1][2][3]. We have added these transductive methods to the related work section.
>
> Secondly, both [4] and our paper utilize semantic information extracted by pre-trained text encoder to improve the performance of few-shot classification. The differences and advantages of our approach will be detailedly discussed in A3 below.
>
> Thank you for reminding us of these related papers. We have cited and discussed them in the revised paper.
>
> **Q2.** In the bi-level optimization, the metric module is updated in the inner loop and then all parameters are updated later in the outer loop. Could the authors provide any reasons or motivation for this training strategy?
>
> **A2.** Thanks for your question. We design this training strategy to make the metric module task-specific. **Intuitively, feature extraction is general for downstream classification tasks. However, image-text alignment, which is conducted by the metric module, is task-specific as each concrete classification task focuses on different parts of features.** Therefore, we make the metric module task-specific by updating it in the inner loop, and updating other modules, such as the feature extractor, in the outer loop. We have added the above discussion to the revised paper.
>
> **Q3.** This paper aims to minimize the gap between the visual and textual modalities by introducing the metric module. It adds degrees of freedom so that there are some learnable parameters for bridging the gap. However, vision-language models are capable of zero-shot inference, therefore the way initializing the "bridge" between two modalities is crucial to the performance. Have the authors considered leveraging category name embedding for initialization [4]? By doing so, two modalities will be aligned automatically.
>
> **A3.** Thanks for your suggestion.
>
> Firstly, we would like to clarify that **the initialization of our method is equivalent to using the category name embedding for initialization**. Specifically, in our implementation, the metric module is represented as a bilinear function $M(z, t) = z^TMt$, where $z$ is the visual feature, $t$ is the textual feature, and $M$ is a square matrix with learnable parameters for bridging the gap. We set $M$ to an identity matrix at the initial state in our implementation. In this scenario, it is equivalent to leveraging the category name embedding for initialization, as $M\cdot t=I\cdot t=t$ and $M(z, t) = z^Tt$. We have added this implementation detail in the revised paper.
>
> Secondly, we would like to highlight that direct alignment leads to poor performance, as described in Figure 1 of our paper. Therefore, we design the bi-level optimization to update the metric module $M$ and make it task-specific.
>
> Thirdly, [4] chooses the pre-trained textual encoder of CoCa, which is a strong vision language model as the language model. However, for language models pre-trained without image-text pairs, the alignment between image and text is more difficult. Our experiments with GloVe and RoBERTa show that our method is also effective with this kind of language model.

---

> ### Author Response · Authors · 2023-11-19
> **Response to Reviewer sXdY (2/2)**
>
> **Q4.** Comparison regarding other state-of-the-art few-shot approaches with established results on CUB or Mini-Imagenet.
>
> **A4.** Thank you for the suggestion. As mentioned in **A1**, it is unfair to compare performance between inductive and transductive FSL methods as they are two different branches of few-shot learning. Regarding the SOTA inductive methods,  we listed these approaches in Table 1 and Table 2 of our paper, which are conducted on CUB and Mini-Imagenet. Notably, our method achieves SOTA performance among them.
>
> **Q5.** Explanation of the motivation or insights of the training strategy, especially why only the metric module is updated in the inner loop and all parameters are updated in the outer loop.
>
> **A5.** As this question is similar to **Q2**, please see **A2** for the explanation of the motivation of the training strategy.
>
> **Q6.** Discussion about the category name initialization and possible comparison or further experiments by adding that strategy to existing approach.
>
> **A6.** Thanks for the suggestion. As explained in **A3**, the initialization of our method is equivalent to using the category name embedding for initialization.

---

> > ### Comment · Reviewer_sXdY · 2023-11-22
> > **Response to the rebuttal**
> >
> > Thank you for your response. The authors have satisfactorily tackled my inquiries. My sole reservation pertains to the novelty aspect, given the authors' assertion that the proposed method mirrors category name initialization.

---

> ### Author Response · Authors · 2023-11-20
> **We would be grateful if you could take a look at the response**
>
> Dear Reviewer sXdY:
>
> We sincerely appreciate your valuable time devoted to reviewing our manuscript. We would like to gently remind you of the **approaching deadline for the discussion phase**. We have diligently addressed the issues you raised in your feedback, providing detailed explanations. For instance, we have clarified the difference between inductive FSL and transductive FSL. Moreover, we have explained the motivation for our bi-level optimization training strategy. Besides, we have made a discussion about the relationship between the initialization using category name embedding and that of our method. Would you kindly take a moment to look at it?
>
> We are very enthusiastic about engaging in more in-depth discussions with you.

---

> ### Author Response · Authors · 2023-11-22
>
> Dear Reviewer sXdY,
>
> As the discussion period approaches its conclusion, **we want to ensure that we have thoroughly addressed all your concerns and that our revised paper fully meets the standards of ICLR**. We would highly value any additional feedback you may provide.
>
> Thank you sincerely for your time and consideration.
>
> Best regards,
> The Authors

---

### Author Response · Authors · 2023-11-19
**A Summary of Paper Updates**

We thank all reviewers for the constructive suggestions, which help make this work more complete. Following their suggestions, we have made the following major updates to the paper:

- Adding references to transductive few-shot learning works to section 2 as reviewers sXdY and Pbba mentioned.

- Adding technical details about the learnable prompt template to section 4.1 as reviewer rhJx requests.

- Adding implementation details about the initialization of the metric module to section 5.1 as reviewer sXdY suggests.

- Adding new experiments to section 5 and appendix, including the ablation study on the visual backbone as reviewer Pbba requests, and experiments on the meta-dataset (including COCO) as reviewers Pbba and 72Kp requests.

- Adding discussions of how our method differs from related works in section 2 and appendix, including zero-shot learning as reviewers rhJx and 72Kp request, previous semantic few-shot learning as reviewers Pbba and 72Kp request, and multimodal contrastive learning as reviewer Pbba requests.

---

### Meta-Review · Area_Chair_dCgs · 2023-12-08

**Metareview:**

This paper introduces a few-shot learning method by benefitting from aligninment of vision and language modalities.
The approach involves using a language branch to generate semantic embeddings for few-shot classes, which are then aligned with visual representations from an image encoder through a learned metric. This process is formulated as a meta-learning problem, with the metric being learned in the inner-loop and the training of vision/language encoders in the outer-loop of optimization.

The paper was reviewed by four experts, who, while recognizing the merit of the developed method, found its technical novelty to be limited. The AC concurs with the reviewers, leading to the conclusion that the paper is not ready for publication at ICLR.

Last but not least, the term metric used in the paper loosely, as a vanilla MLP cannot by default realize a metric (eg, a metric is symmetric).
The authors are advised to revise either the text or the implementation of their algorithm to ensure rigorous adherence to the concept of a metric.

**Justification For Why Not Higher Score:**

The paper received four reject scores (5,3,3,3), all pointing out the lack of novelty. This is indeed the case and hence I believe the paper should be rejected.

**Justification For Why Not Lower Score:**

NA

---

### Decision · Program_Chairs · 2024-01-16

Reject